# DENSE: Data-Free One-Shot Federated Learning

**Jie Zhang**[1*]   **Chen Chen**[1*]   **Bo Li**[2‡]   **Lingjuan Lyu**[3‡]
**Shuang Wu**[2]   **Shouhong Ding**[2]   **Chunhua Shen**[1]   **Chao Wu**[1‡]

[1]Zhejiang University   [2]Youtu Lab, Tencent   [3] Sony AI

{zj_zhangjie, chao.wu, cc33}@zju.edu.cn, Lingjuan.Lv@sony.com
{libraboli, calvinwu, ericshding}@tencent.com, chunhua@me.com

## Abstract

One-shot Federated Learning (FL) has recently emerged as a promising approach, which allows the central server to learn a model in a single communication round. Despite the low communication cost, existing one-shot FL methods are mostly impractical or face inherent limitations, *e.g.*, a public dataset is required, clients' models are homogeneous, and additional data/model information need to be uploaded. To overcome these issues, we propose a novel two-stage **D**ata-fre**E** o**N**e-**S**hot federated l**E**arning (DENSE) framework, which trains the global model by a data generation stage and a model distillation stage. DENSE is a practical one-shot FL method that can be applied in reality due to the following advantages: (1) DENSE requires no additional information compared with other methods (except the model parameters) to be transferred between clients and the server; (2) DENSE does not require any auxiliary dataset for training; (3) DENSE considers model heterogeneity in FL, *i.e.*, different clients can have different model architectures. Experiments on a variety of real-world datasets demonstrate the superiority of our method. For example, DENSE outperforms the best baseline method Fed-ADI by 5.08% on CIFAR10 dataset.

## 1   Introduction

Deep neural networks (DNNs) have recently gained popularity as a powerful tool for advancing artificial intelligence in both established and emerging fields [25, 22, 13, 9, 8, 51, 52, 58, 5, 4, 16]. Federated learning (FL) [42] has emerged as a promising learning paradigm which allows multiple clients to collaboratively train a global model without exposing their private training data. In FL, each client trains a local model on its own data and is required to periodically share its high-dimensional model parameters with a central server. Recent years, FL has shown its potential to facilitate real-world applications in many fields, including medical image analysis [36, 6], recommender systems [34, 38], natural language processing [63, 46] and computer vision [27, 26].

The original FL framework requires participants to communicate frequently with the central server in order to exchange models. In real-world FL, such high communication cost may be intolerable and impractical. Reducing the communication cost between clients and the server is desired both for system efficiency and to support the privacy goals of federated learning [40, 41]. Recent research proposed some common methods to reduce communication costs, *e.g.*, utilize multiple local updates [18], employ compression techniques [48], and one-shot FL [10]. Among them, **one-shot FL** is

---

[*]Both authors contributed equally to this work. Work done during Jie Zhang's internship at Tencent Youtu Lab and partly done at Sony AI.

[*]Work is completed during Chen Chen's internship at Sony AI.

[‡]Corresponding author.

36th Conference on Neural Information Processing Systems (NeurIPS 2022).

a promising solution which only allows one communication round. There are several motivations behind one-shot FL: **1)** First of all, multi-round training is not practical in some scenarios such as model markets [45], in which users can only buy the pre-trained models from the market without any real data. **2)** Furthermore, frequent communication poses a high risk of being attacked. For instance, frequent communication can be easily intercepted by attackers, who can launch man-in-the-middle attacks [47] or even reconstruct the training data from gradients [54]. In this way, one-shot FL can reduce the probability of being intercepted by malicious attackers due to the one-round property. Thus, in this paper, we mainly focus on one-shot FL.

However, existing one-shot FL studies [10, 30, 62, 7] are still hard to apply in real-world applications, due to impractical settings. For example, Guha *et al.* [10] and Li *et al.* [30] involved a public dataset for training, which may be impractical in very sensitive scenarios such as the biomedical domains. Zhu *et al.* [62] adopted dataset distillation [50] in one-shot FL, but they need to send distilled data to the central server, which causes additional communication cost and potential privacy leakage. Dennis *et al.* [7] utilized cluster-based method in one-shot FL, which requires to upload the cluster means to the server, causing additional communication cost. Additionally, none of these methods consider model heterogeneity, *i.e.*, different clients have different model architectures [31], which is very common in practical scenarios. For instance, in model market, models sold by different sellers are likely to be heterogeneous. Besides, when several medical institutions participate in FL, they may need to design their own model to meet distinct specifications. Therefore, developing a practical one-shot FL method is in urgent need.

In this work, we propose a novel two-stage **D**ata-fre**E** o**N**e-**S**hot federated l**E**arning (DENSE) framework, which trains the global model by a data generation stage and a model distillation stage. In the first stage, we utilize the ensemble models (*i.e.*, ensemble of local models uploaded by clients) to train a generator, which can generate synthetic data for training in the second stage. In the second stage, we distill the knowledge of the ensemble models to the global model. In contrast to traditional FL methods based on FedAvg [42], our method does not require averaging of model parameters, thus it can support heterogeneous models, *i.e.*, clients can have different model architectures. In summary, our main contributions are summarized as follows:

- We propose a novel data-free one-shot FL framework named DENSE, which consists of two stages. In the first stage, we train a generator that considers *similarity*, *stability*, and *transferability* at the same time. In the second stage, we use the ensemble models and the data generated by the generator to train a global model.

- The setting of DENSE is practical in the following aspects. First, DENSE requires no additional information (except the model parameters) to be transferred between clients and the server; Second, DENSE does not require any auxiliary dataset for training; Third, DENSE considers model heterogeneity, *i.e.*, different clients can have different model architectures.

- DENSE is a compatible approach, which can be combined with any local training techniques to further improve the performance of the global model. For instance, we can adopt LDAM [1] to train the clients' local models, and improve the accuracy of the global model (refer to Section 2.3 and Section 3.2).

- Extensive experiments on various datasets verify the effectiveness of our proposed DENSE. For example, DENSE outperforms the best baseline method Fed-ADI [55] by 5.08% on CIFAR10 dataset.

## 2 Data-Free One-Shot Federated Learning

### 2.1 Framework Overview

To tackle the problems in recent one-shot FL methods as mentioned in Sec. 1, we propose a novel method named DENSE, which conducts one-shot FL without the need to share additional information or rely on any auxiliary dataset, while considering model heterogeneity. To simulate real-world applications, we consider a more challenging yet practical setting where the data on each client are not independent and identically distributed (non-IID).

The illustration of the learning procedure is demonstrated in Figure 1, and the whole training process of DENSE is shown in Algorithm 1. After clients upload their local models to the server, the server

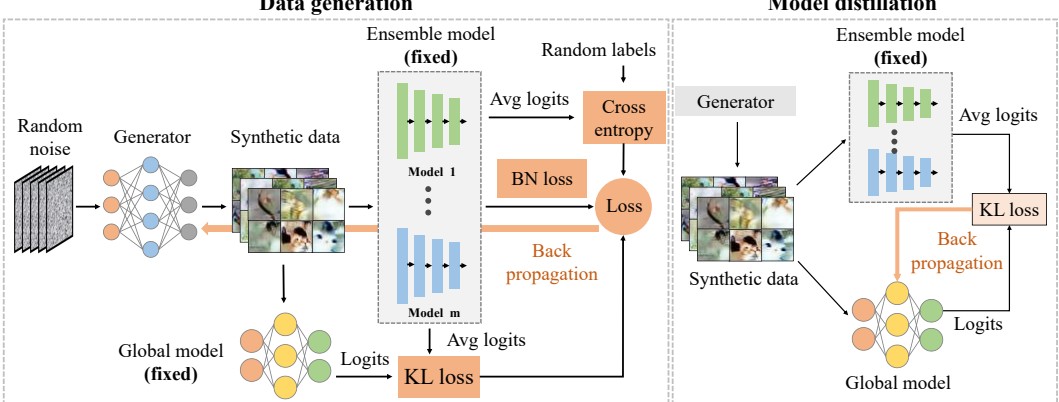

Figure 1: An illustration of training process of DENSE on the server, which consists of two stages: (1) In data generation stage, we train an auxiliary generator that considers similarity, stability, and transferability at the same time; (2) In model distillation stage, we distill the knowledge of the ensemble models and transfer to the global model. Note that the fixed global model is used as an additional discriminator in the divergence loss $\mathcal{L}_{div}$.

trains a global model with DENSE in two stages. In the data generation stage (first stage), we train an auxiliary generator that can generate synthetic data by the ensemble models, *i.e.*, ensemble of local models uploaded by clients. In the model distillation stage (second stage), we use the ensemble models and the synthetic data (generated by the generator) to train the global model.

## 2.2 Data Generation

In the first stage, we aim to train a generator to generate synthetic data. Specifically, given the ensemble of well-trained models uploaded by clients, our goal is to train a generator that can generate data that have similar distribution to the training data of clients. In addition, we aim not to leak private information from our generated data, *i.e.*, attackers are not able to predict any sensitive information of clients from the generated data. Recent work [35] generated data by utilizing a pre-trained generative adversarial network (GAN). However, such a method is unable to generate data as the pre-trained GAN is trained on public datasets, which is likely to have different data distribution from the training data of clients. Moreover, we need to consider model heterogeneity, which makes the problem more complicated.

To solve these issues, we propose to train a generator that considers *similarity*\*, *stability*, and *transferability*. The data generation process is shown in line 8 to 11 in Algorithm 1. In particular, given a random noise $\mathbf{z}$ (generated from a standard Gaussian distribution) and a random one-hot label $\mathbf{y}$ (generated from a uniform distribution), the generator $G(\cdot)$ aims to generate a synthetic data $\hat{\mathbf{x}} = G(\mathbf{z})$ such that $\hat{\mathbf{x}}$ is similar to the training data (with label $\mathbf{y}$) of clients.

**Similarity.** First, we need to consider the similarity between synthetic data $\hat{\mathbf{x}}$ and the training data. Since we are unable to access the training data of clients, we cannot compute the similarity between the synthetic data and the training data directly. Instead, we first compute the average logits (*i.e.*, outputs of the last fully connected layer) of $\hat{\mathbf{x}}$ computed by the ensemble models.

$$D(\hat{\mathbf{x}}; \{\boldsymbol{\theta}^k\}_{k=1}^m) = \frac{1}{m} \sum_{k \in \mathcal{C}} f^k\left(\hat{\mathbf{x}}; \boldsymbol{\theta}^k\right), \tag{1}$$

where $m = |\mathcal{C}|$, and $D(\hat{\mathbf{x}}; \{\boldsymbol{\theta}^k\}_{k=1}^m)$ is the average logits of $\hat{\mathbf{x}}$, $\boldsymbol{\theta}^k$ is the parameter of the $k$-th client. And $f^k\left(\hat{\mathbf{x}}; \boldsymbol{\theta}^k\right)$ is the prediction function of client $k$ that outputs the logits of $\hat{\mathbf{x}}$ given parameter $\boldsymbol{\theta}^k$. For simplicity, we use $D(\hat{\mathbf{x}})$ to denote $D(\hat{\mathbf{x}}; \{\boldsymbol{\theta}^k\}_{k=1}^m)$ in the rest of the paper.

---

\*Note that the ideal synthetic data should be visually distinct from the real data for visual privacy, but similar in distribution for utility.

Then, we minimize the average logits and the random label $y$ with the following cross-entropy (CE) loss.

$$\mathcal{L}_{CE}(\hat{\mathbf{x}}, \mathbf{y}; \boldsymbol{\theta}_G) = CE(D(\hat{\mathbf{x}}), \mathbf{y}), \tag{2}$$

It is expected that the synthetic images can be classified into one particular class with a high probability by the ensemble models. In fact, during the training phase, the loss between $D(\hat{\mathbf{x}})$ and $\mathbf{y}$ can easily reduce to almost 0, which indicates the synthetic data matches the ensemble models perfectly. Moreover, we do not directly compute the similarity between the synthetic data and the training data, which can reduce the probability of leaking sensitive information of the clients.

However, by utilizing only the CE loss, we cannot achieve a high performance (please refer to Section 3.2 for detail). We conjecture this is because the ensemble models are trained on non-IID data, the generator may be unstable and trapped into sub-optimal local minima or overfit to the synthetic data [49, 32].

**Stability.** Second, to improve the stability of the generator, we propose to add an additional regularization to stabilize the training. In particular, we utilize the Batch Normalization (BN) loss to make the synthetic data conform with the batch normalization statistics [55].

$$\mathcal{L}_{BN}(\hat{\mathbf{x}}; \boldsymbol{\theta}_G) = \frac{1}{m} \sum_{k \in \mathcal{C}} \sum_{l} \left( \|\mu_l(\hat{\mathbf{x}}) - \mu_{k,l}\| + \|\sigma_l^2(\hat{\mathbf{x}}) - \sigma_{k,l}^2\| \right), \tag{3}$$

where $\mu_l(\hat{\mathbf{x}})$ and $\sigma_l^2(\hat{\mathbf{x}})$ are the batch-wise mean and variance estimates of feature maps corresponding to the $l$-th BN layer of the generator $G(\cdot)^\dagger$, $\mu_{k,l}$ and $\sigma_{k,l}^2$ are the mean and variance of the $l$-th BN layer [17] of $f^k(\cdot)$. The BN loss minimizes the distance between the feature map statistics of the synthetic data and the training data of clients. As a result, the synthetic data can have a similar distribution to the training data of clients, no matter if the data is non-IID or IID.

**Transferability.** By utilizing the CE loss and BN loss, we can train a generator that can generate synthetic data, but we observed that the synthetic data are likely to be far away from the decision boundary (of the ensemble models), which makes the ensemble models (teachers) hard to transfer their knowledge to the global model (student). We illustrate the observation in the left panel of Figure 2. S and T are the decision boundaries of the global model (the detail of the global model is introduced in Section 2.3) and ensemble models respectively. The essence of knowledge distillation is transferring the information of decision boundary from the teacher model to the student model [12]. We aim to learn the decision boundary of global model and have a high classification accuracy on the real test data (blue diamonds). However, the generated synthetic data (red circles) are likely to be on the same side of the two decision boundaries and unhelpful to the transfer of knowledge [12]. To solve this problem, we

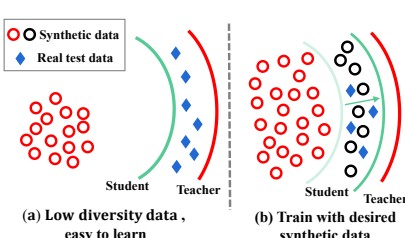

Figure 2: The illustration of generated data and decision boundary of ensemble models (teachers) and global model (student). **Left panel:** Synthetic data (red circles) are far away from the decision boundary, which is less helpful to the transfer of knowledge. **Right panel:** By utilizing our boundary support loss, we can generate more synthetic data near the decision boundaries (black circles), which helps the student better learn the decision boundary of the teacher.

argue to generate more synthetic data that fall between the decision boundaries of the ensemble models and the global model. We illustrate our idea in the right panel of Figure 2. Red circles are synthetic data on the same side of the decision boundary, which are less helpful in learning the global model. Black circles are synthetic data between the decision boundaries, *i.e.*, the global model and the ensemble models have different predictions on these data. Black circles can help the global model better learn the decision boundary of the ensemble models.

Motivated by the above observations, we introduce a new boundary support loss, which urges the generator to generate more synthetic data between the decision boundaries of the ensemble models and the global model. We divide the synthetic data into 2 sets: (1) the global model and the ensemble

---

$^\dagger$We assume the input is a batch of data.

models have the same predictions on data in the first set $(\arg\max_c D^{(c)}(\hat{\mathbf{x}}) = \arg\max_c f_S^{(c)}(\hat{\mathbf{x}}; \boldsymbol{\theta}_S))$; (2) different predictions on data in the second set $(\arg\max_c D^{(c)}(\hat{\mathbf{x}}) \neq \arg\max_c f_S^{(c)}(\hat{\mathbf{x}}; \boldsymbol{\theta}_S))$, where $D^{(c)}(\hat{\mathbf{x}})$ and $f_S^{(c)}(\hat{\mathbf{x}}; \boldsymbol{\theta}_S)$ are the logits for the $c$-th label of the ensemble models and the global model respectively. The data in the first set are on the same side of those two decision boundaries (red circles in Figure 2) while the data in the second set (black circles in Figure 2) are between the decision boundaries of the ensemble models and the global model. We maximize the differences of predictions of the global model and the ensemble models on data in the second set with Kullback-Leibler divergence loss as follows.

$$\mathcal{L}_{div}(\hat{\mathbf{x}}; \boldsymbol{\theta}_G) = -\omega KL\left(D(\hat{\mathbf{x}}), f_S(\hat{\mathbf{x}}; \boldsymbol{\theta}_S)\right), \tag{4}$$

where $KL(\cdot, \cdot)$ denotes the Kullback-Leibler (KL) divergence loss, $\omega = \mathbb{1}(\arg\max_c D^{(c)}(\hat{\mathbf{x}}) \neq \arg\max_c f_S^{(c)}(\hat{\mathbf{x}}; \boldsymbol{\theta}_S))$ outputs 0 for data in the first set and 1 for data in the second set, and $\mathbb{1}(a)$ is the indicator function that outputs 1 if $a$ is true and outputs 0 if $a$ is false. By maximizing the KL divergence loss, the generator can generate more synthetic data that are more helpful to the model distillation stage (refer to Section 2.3 for detail) and further improve the transferability of the ensemble models.

By combining the above losses, we can obtain the generator loss as follows,

$$\mathcal{L}_{gen}(\hat{\mathbf{x}}, \mathbf{y}; \boldsymbol{\theta}_G) = \mathcal{L}_{CE}(\hat{\mathbf{x}}, \mathbf{y}; \boldsymbol{\theta}_G) + \lambda_1 \mathcal{L}_{BN}(\hat{\mathbf{x}}; \boldsymbol{\theta}_G) + \lambda_2 \mathcal{L}_{div}(\hat{\mathbf{x}}; \boldsymbol{\theta}_G), \tag{5}$$

where $\lambda_1$ and $\lambda_2$ are scaling factors for the losses.

---

**Algorithm 1** Training process of DENSE

---

**Input:** Number of client $m$, clients' local models $\{f^1(), \cdots, f^m()\}$, generator $G(\cdot)$ with parameter $\boldsymbol{\theta}_G$, learning rate of the generator $\eta_G$, number of training rounds $T_G$ for generator in each epoch, global model $f_S()$ with parameter $\boldsymbol{\theta}_S$, learning rate of the global model $\eta_S$, global model training epochs $T$, and batch size $b$.

    **for** each client $k \in \mathcal{C}$ **in parallel do**
        $\theta^k \leftarrow$ **LocalUpdate**$(k)$
    **end for**
    Initialize parameter $\boldsymbol{\theta}_G$ and $\boldsymbol{\theta}_S$
    **for** epoch$=1, \cdots, T$ **do**
        Sample a batch of noises and labels $\{\mathbf{z}_i, \mathbf{y}_i\}_{i=1}^b$
        // data generation stage
        **for** $j = 1, \cdots, T_G$ **do**
            Generate $\{\hat{\mathbf{x}}_i\}_{i=1}^b$ with $\{\mathbf{z}_i\}_{i=1}^b$ and $G(\cdot)$
            $\boldsymbol{\theta}_G \leftarrow \boldsymbol{\theta}_G - \eta_G \frac{1}{b} \sum_{i=1}^b \nabla_{\boldsymbol{\theta}_G} \ell_{gen}(\hat{\mathbf{x}}_i, \mathbf{y}_i; \boldsymbol{\theta}_G)$
        **end for**
        // model distillation stage
        Generate $\{\hat{\mathbf{x}}_i\}_{i=1}^b$ with $\{\mathbf{z}_i\}_{i=1}^b$ and $G(\cdot)$
        $\boldsymbol{\theta}_S = \boldsymbol{\theta}_S - \eta_S \frac{1}{b} \sum_{i=1}^b \nabla_{\boldsymbol{\theta}_S} \ell_{dis}(\hat{\mathbf{x}}_i; \boldsymbol{\theta}_S)$
    **end for**

---

Note that the generated synthetic data have similar features but different from the training data (of clients), which reduces the probability of leaking sensitive information of clients. More discussions of the privacy issues are in Section 3.3.3.

## 2.3 Model Distillation

In the second stage, we train the global model with the generator (discussed in the previous section) and the ensemble models. Previous research [60, 35] showed that model ensemble provides a general method for improving the accuracy and stability of learning models. Motivated by [60], we propose to use the ensemble models as a teacher to train a student (global) model. A straightforward method is to obtain the global model by aggregating the parameters of all client models (*e.g.*, by FedAvg [42]). However, in real-world applications, clients are likely to have different model architectures [44], making FedAvg useless. Moreover, since the data in different clients are non-IID, FedAvg cannot deliver a good performance or even diverge [59, 32].

To this end, we follow [35] to distill the knowledge of the ensemble models to the global model by minimizing the predictions between the ensemble models (teacher) and the global model (student) on the same synthetic data. The model distillation process is shown in line 13 to 14 in Algorithm 1. First, we compute the average logits of the synthetic data according to Eq. (1), *i.e.*, $D(\hat{\mathbf{x}}) = \frac{1}{m}\sum_{k\in\mathcal{C}} f^k\left(\hat{\mathbf{x}};\boldsymbol{\theta}^k\right)$. In contrast to traditional aggregation methods (*e.g.*, FedAvg) that are unable to aggregate heterogeneous models, averaging logits can be easily applied to both heterogeneous and homogeneous FL systems.

Then, we use the average logits to distill the knowledge of the ensemble models by minimizing the following objective function.

$$\mathcal{L}_{dis}(\hat{\mathbf{x}};\boldsymbol{\theta}_S) = KL\left(D(\hat{\mathbf{x}}), f_S(\hat{\mathbf{x}};\boldsymbol{\theta}_S)\right). \tag{6}$$

By minimizing the KL loss, we can train a global model with the knowledge of the ensemble models and the synthetic data regardless of data and model heterogeneity.

Note that DENSE has no restriction on the clients' local models, *i.e.*, clients can train models with arbitrary techniques. Thus, DENSE is a compatible approach, which can be combined with any local training techniques to further improve the performance of the global model. We further discuss the combination of local training techniques in Section 3.2.

**Discussions on privacy-preserving** Research [15] has shown that it is possible to launch an attack where a malicious user uses GANs to recreate samples of another participant's private datasets. Besides, in FL, exchanging models between the server and clients can result in potential privacy leakage. Note that our method prohibits the generator from seeing the real data directly, and there is only one communication round, which reduces the risk of privacy leakage. In addition, we display our generated images in Section 3.3.3, which does not directly reveal the information of real data. Several existing privacy-preserving methods can be incorporated into our framework to better protect clients from adversaries [37, 20]. We leave this as our future work.

**Discussions on Knowledge Distillation in FL** In traditional FL frameworks, all users have to agree on the specific architecture of the global model. To support model heterogeneity, Li *et al.* [28] proposed a new federated learning framework that enables participants to independently design their models by knowledge distillation [14]. With the use of a proxy dataset, knowledge distillation alleviates the model drift issue induced by non-IID data. However, the requirement of proxy data renders such a method impractical for many applications, since a carefully designed dataset is not always available on the server. Data-free knowledge distillation is a promising approach, which can transfer knowledge of a teacher model to a student model without any real data [2, 55]. Lin *et al.* [35] proposed data-free ensemble distillation for model fusion through synthetic data in each communication round, which requires high communication costs and computational costs. However, in this paper, we are more concerned with obtaining a good global model through only one round of communication in cases of heterogeneous models, which is more challenging and practical. Zhu *et al.* [64] also proposed a data-free knowledge distillation approach for FL, which learns a generator derived from the prediction of local models. However, the learned generator is later broadcasted to all clients, and then clients need to send their generators to the server, which increases the communication burden. More seriously, the generator has direct access to the local data (the generator can easily remember the training samples [39]), which can cause privacy concerns. As the generator used in our method is always stored in the central server, it never sees any real local data.

## 3 Experiments

### 3.1 Experimental Setup

#### 3.1.1 Datasets

Our experiments are conducted on the following 6 real-world datasets: MNIST [24], FMNIST [53], SVHN [43], CIFAR10 [21], CIFAR100 [21], and Tiny-ImageNet [23].MNIST dataset contains binary images of handwritten digits. There are 60,000 training images and 10,000 testing images in MNIST dataset. CIFAR10 dataset consists of 60,000 32x32 color images in 10 classes, with 6,000 images per class. There are 50,000 training images and 10,000 test images in CIFAR10 dataset. CIFAR100 dataset is similar to CIFAR10 dataset, except it has 100 classes containing 600 images each. There are 500 training images and 100 testing images per class. Tiny-ImageNet contains 100000 images of 200 classes (500 for each class) downsized to 64×64 colored images. Each class has 500 training images, 50 validation images and 50 test images.

Table 1: Accuracy of different methods across $\alpha = \{0.1, 0.3, 0.5\}$ on different datasets.

| Dataset | MNIST | | | FMNIST | | | CIFAR10 | | | SVHN | | | CIFAR100 | | | Tiny-ImageNet | | |
|---|---|---|---|---|---|---|---|---|---|---|---|---|---|---|---|---|---|---|
| Method | $\alpha$=0.1 | $\alpha$=0.3 | $\alpha$=0.5 | $\alpha$=0.1 | $\alpha$=0.3 | $\alpha$=0.5 | $\alpha$=0.1 | $\alpha$=0.3 | $\alpha$=0.5 | $\alpha$=0.1 | $\alpha$=0.3 | $\alpha$=0.5 | $\alpha$=0.1 | $\alpha$=0.3 | $\alpha$=0.5 | $\alpha$=0.1 | $\alpha$=0.3 | $\alpha$=0.5 |
| FedAvg | 48.24 | 72.94 | 90.55 | 41.69 | 82.96 | 83.72 | 23.93 | 27.72 | 43.67 | 31.65 | 61.51 | 56.09 | 4.58 | 11.61 | 12.11 | 3.12 | 10.46 | 11.89 |
| FedDF | 60.15 | 74.01 | 92.18 | 43.58 | 80.67 | 84.67 | 40.58 | 46.78 | 53.56 | 49.13 | 73.34 | 73.98 | 28.17 | 30.28 | 36.35 | 15.34 | 18.22 | 27.43 |
| Fed-DAFL | 64.38 | 74.18 | 93.01 | 47.14 | 80.59 | 84.02 | 47.34 | 53.89 | 58.59 | 53.23 | 76.56 | 78.03 | 28.89 | 34.89 | 38.19 | 18.38 | 22.18 | 28.22 |
| Fed-ADI | 64.13 | 75.03 | 93.49 | 48.49 | 81.15 | 84.19 | 48.59 | 54.68 | 59.34 | 53.45 | 77.45 | 78.85 | 30.13 | 35.18 | 40.28 | 19.59 | 25.34 | 30.21 |
| DENSE (ours) | 66.61 | 76.48 | 95.82 | 50.29 | 83.96 | 85.94 | 50.26 | 59.76 | 62.19 | 55.34 | 79.59 | 80.03 | 32.03 | 37.32 | 42.07 | 22.44 | 28.14 | 32.34 |

### 3.1.2 Data partition

To simulate real-world applications, we use Dirichlet distribution to generate non-IID data partition among clients [56, 29]. In particular, we sample $p_k \sim Dir(\alpha)$ and allocate a $p_k^i$ proportion of the data of class $k$ to client $i$. By varying the parameter $\alpha$, we can change the degree of imbalance. A small $\alpha$ generates highly skewed data. We set $\alpha = 0.5$ as default.

### 3.1.3 Baselines

To ensure fair comparisons, we neglect the comparison with methods that require to download auxiliary models or datasets, such as FedBE [3] and FedGen [64]. Moreover, since there is only one communication round, aggregation methods that are based on regularization have no effect. Thus, we also omit the comparison with these regularization-based methods, *e.g.*, FedProx [31], FedNova [49], and Scaffold [18]. Instead, we compare our proposed DENSE with FedAvg [42] and FedDF [35]. Furthermore, since DENSE is a data-free method, we derive some baselines from prevailing data-free knowledge distillation methods, including: 1) DAFL [2], a novel data-free learning framework based on generative adversarial networks; 2) ADI [55], an image synthesizing method that utilizes the image distribution to train a deep neural network without real data. We apply these methods to one-shot FL, and name these two baselines as Fed-DAFL and Fed-ADI.

### 3.1.4 Settings

For clients' local training, we use the SGD optimizer with momentum=0.9 and learning rate=0.01. We set the batch size $b = 128$, the number of local epochs $E = 200$, and the client number $m = 5$. Following the setting of [2], we train the auxiliary generator $G(\cdot)$ with a deep convolutional network. We use Adam optimizer with learning rate $\eta_G = 0.001$. We set the number of training rounds in each epoch as $T_G = 30$, and set the scaling factor $\lambda_1 = 1$ and $\lambda_2 = 0.5$. For the training of the server model $f_S()$, we use the SGD optimizer with learning rate $\eta_S = 0.01$ and momentum=0.9. The number of epochs for distillation $T = 200$. All baseline methods use the same setting as ours.

## 3.2 Results

### 3.2.1 Evaluation on real-world datasets

To evaluate the effectiveness of our method, we conduct experiments under different non-IID settings by varying $\alpha = \{0.1, 0.3, 0.5\}$ and report the performance on different datasets and different methods in Table 1. The results show that: (1) Our DENSE achieves the highest accuracy across all datasets. In particular, DENSE outperforms the best baseline method Fed-ADI [55] by 5.08% when $\alpha = 0.3$ on CIFAR10 dataset. (2) FedAvg has the worst performance, which implies that directly averaging the model parameters cannot achieve a good performance under non-IID setting in one-shot FL. (3) As $\alpha$ becomes smaller (*i.e.*, data become more imbalanced), the performance of all methods decrease significantly, which shows that all methods suffer

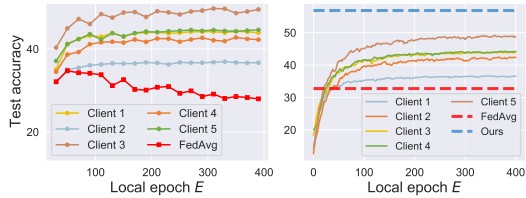

Figure 3: **Left panel:** Accuracy of FedAvg and clients' local models across different local training epochs $E = \{20, 40, 60, \cdots, 400\}$. **Right panel:** The accuracy curve for local training. The dotted lines represent the best results of two one-shot FL methods (FedAvg and DENSE). Our DENSE outperforms FedAvg and local models consistently.

Table 2: Accuracy comparisons across heterogeneous client models on CIFAR10. There are five clients in total, and each client has a personalized model.

| Model | Client | | | | | Server (ResNet-18) | | | |
|---|---|---|---|---|---|---|---|---|---|
| | ResNet-18 | CNN1 | CNN2 | WRN-16-1 | WRN-40-1 | FedDF | Fed-DAFL | Fed-ADI | **DENSE (ours)** |
| $\alpha$=0.1 | 40.83 | 33.67 | 35.21 | 27.73 | 32.93 | 42.35 | 43.12 | 44.63 | **49.76** |
| $\alpha$=0.3 | 51.49 | 52.78 | 44.96 | 47.35 | 37.24 | 52.72 | 57.72 | 58.96 | **63.25** |
| $\alpha$=0.5 | 59.96 | 58.67 | 54.28 | 53.39 | 58.14 | 60.05 | 61.56 | 63.24 | **67.42** |

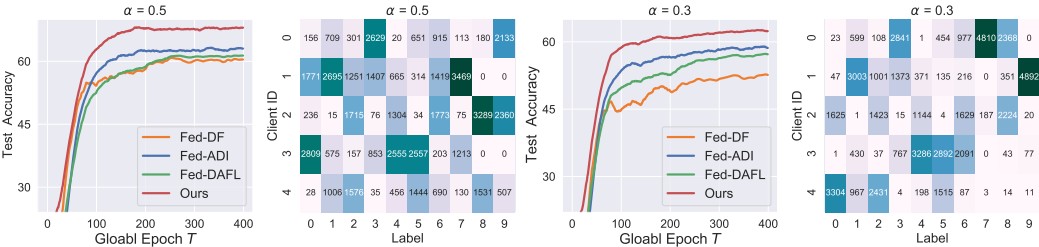

Figure 4: Visualization of the test accuracy and data distribution for CIFAR10 with $\alpha = \{0.3, 0.5\}$.

from highly skewed data. Even under highly skewed setting, DENSE still significantly outperforms other methods, which further demonstrates the superiority of our proposed method.

### 3.2.2 Impact of model distillation

We show the impact of model distillation by comparing with FedAvg. We first conduct one-shot FL and use FedAvg to aggregate the local models. We show the results of the global model and clients' local models across different local training epochs $E = \{20, 40, 60, \cdots, 400\}$ in the left panel of Figure 3. The global model achieves the best performance (test accuracy=34%) when $E = 40$, while a larger value of $E$ can cause the model to degrade even collapse. This result can be attributed to the inconsistent optimization objectives with non-IID data [49], which leads to weight divergence [61]. Then, we show the results of one-shot FL when $E = 400$ and report the performance of FedAvg and DENSE in the right panel of Figure 3. We also plot the performance of clients' local models. DENSE outperforms each client's local model while FedAvg underperforms each client's local model. This validates that model distillation can enhance training while directly aggregating is harmful to the training under non-IID setting in one-shot FL.

### 3.2.3 Results in heterogeneous FL

Note that our proposed DENSE can support heterogeneous models. We apply five different CNN models on CIFAR10 dataset with Dirichlet distribution $\alpha = \{0.1, 0.3, 0.5\}$. The heterogeneous models include: 1) one ResNet-18 [11], 2) two small CNNs: CNN1 and CNN2; 3) two Wide-ResNets (WRN) [57]: WRN-16-1 and WRN-40-1. For knowledge distillation, we use ResNet-18 as the server's global model. Detailed architecture information of the given deep networks can be found in Appendix. Table 2 evaluates all methods in heterogeneous one-shot FL under practical non-IID data settings. We omit the results for FedAvg as FedAvg does not support heterogeneous models. We remark that FL under both the non-IID data distribution and different model architecture setting is a quite challenging task. Even under this setting, our DENSE still significantly outperforms other baselines. In addition, we report the accuracy curve of global distillation. As shown in Figure 4, our method outperforms other baselines by a large margin.

## 3.3 Analysis of Our Method

### 3.3.1 Impact of the number of clients

Furthermore, we evaluate the performance of these methods on CIFAR10 and SVHN datasets by varying the number of clients $m = \{5, 10, 20, 50, 100\}$. According to [33], the server can become a bottleneck when the number of clients is very large, we are also concerned with the model performance

Table 3: Accuracy across different number of clients $m = \{5, 10, 20, 50, 100\}$ on CIFAR10 and SVHN datasets.

| Dataset | CIFAR10 | | | | | SVHN | | | | |
|---|---|---|---|---|---|---|---|---|---|---|
| $m$ | FedAvg | FedDF | Fed-DAFL | Fed-ADI | **DENSE (ours)** | FedAvg | FedDF | Fed-DAFL | Fed-ADI | **DENSE (ours)** |
| 5 | 43.67 | 53.56 | 55.46 | 58.59 | **62.19** | 56.09 | 73.98 | 78.03 | 78.85 | **80.03** |
| 10 | 38.29 | 54.44 | 56.34 | 57.13 | **61.42** | 45.34 | 62.12 | 63.34 | 65.45 | **67.57** |
| 20 | 36.03 | 43.15 | 45.98 | 46.45 | **52.71** | 47.79 | 60.45 | 62.19 | 63.98 | **66.42** |
| 50 | 37.03 | 40.89 | 43.02 | 44.47 | **48.47** | 36.53 | 51.44 | 54.23 | 57.35 | **59.27** |
| 100 | 33.54 | 36.89 | 37.55 | 36.98 | **43.28** | 30.18 | 46.58 | 47.19 | 48.33 | **52.48** |

Table 4: Performance analysis of DENSE+LDAM.

| Dataset | CIFAR10 | | | SVHN | | |
|---|---|---|---|---|---|---|
| Method | $\alpha$=0.1 | $\alpha$=0.3 | $\alpha$=0.5 | $\alpha$=0.1 | $\alpha$=0.3 | $\alpha$=0.5 |
| DENSE | 50.26 | 59.76 | 62.19 | 55.34 | 79.59 | 80.03 |
| DENSE+LDAM | **57.24** | **63.13** | **64.76** | **58.04** | **81.28** | **81.77** |

when $m$ increases. Table 3 shows the results of different methods across different $m$. The accuracy of all methods decreases as the number of clients $m$ increases, which is consistent with observations in [33, 42]. Even though the number of clients can affect the performance of one-shot FL, our method still outperforms other baselines. The increasing number of clients poses new challenges for ensemble distillation, which we leave for future investigation.

### 3.3.2 Combination with imbalanced learning

The accuracy of federated learning reduces significantly with non-IID data, which has been broadly discussed in recent studies [29, 49]. Additionally, previous studies [1, 19] have demonstrated their superiority on imbalanced data. The combination of our method with these techniques to address imbalanced local data can lead to a more effective FL system. For example, by using LDAM [1] in clients' local training, we can mitigate the impact of data imbalance, and thereby build a more powerful ensemble model. We compare the performance of the original DENSE and DENSE combined with LDAM (DENSE+LDAM) across $\alpha = \{0.1, 0.3, 0.5\}$ on CIFAR10 and SVHN datasets.

As demonstrated in Table 4, DENSE+LDAM can significantly improve the performance, especially for highly skewed non-IID data (i.e. $\alpha = 0.1$). To help understand the performance gap and data skewness, in Figure 5, we visualize the accuracy curve and data distribution of CIFAR10 ($\alpha$=0.1) in the left panel and right panel respectively. The number in the right panel stands for the number of examples associated with the corresponding label in one particular client. These figures imply that significant improvement can be achieved by combining DENSE with LDAM on highly skewed data.

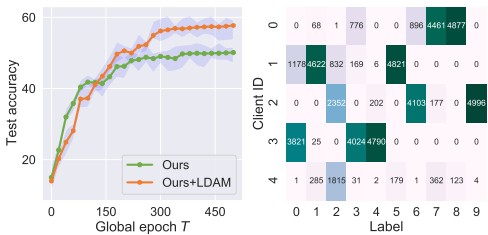

Figure 5: **Left panel:** Accuracy curves of DENSE and DENSE+LDAM. **Right panel:** Data distribution of different clients for CIFAR10 dataset ($\alpha$=0.1).

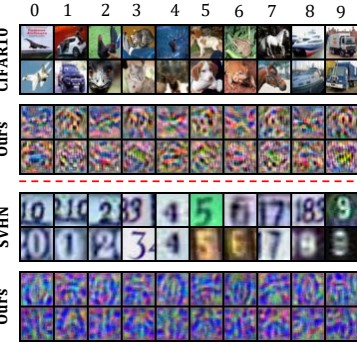

Figure 6: Visualization of synthetic data on CIFAR10 and SVHN datasets.

### 3.3.3 Visualization of synthetic data

To compare the synthetic data with the training data, we visualize the synthetic data on CIFAR10 and SVHN datasets in Figure 6. As shown in the figure, the first / third row is the original data of CIFAR10 / SVHN dataset, and the second / last row is the synthetic data generated by the model trained on CIFAR10 / SVHN dataset. The synthetic data are not similar to the original data, which can effectively reduce the probability of leaking sensitive information of clients. Note that although the synthetic data look much different from the original data, our method still achieves a higher performance than other baseline methods by training with these synthetic data (as shown in Table 1). Note that the ideal synthetic data should be visually distinct from the real data.

### 3.3.4 Extend to multiple rounds

We also extend DENSE to multi-round FL to test its effectiveness, *i.e.*, there are multiple communication rounds between clients and server. Table 5 demonstrates the results of DENSE across different communication rounds $T_c = \{1, 2, 3, 4, 5\}$ on CIFAR10 and SVHN datasets. The local training epoch is fixed as $E = 10$. The performance of

Table 5: Accuracy for multiple communication rounds.

| Dataset | CIFAR10 | | | SVHN | | |
|---|---|---|---|---|---|---|
| Communication rounds | $\alpha$=0.1 | $\alpha$=0.3 | $\alpha$=0.5 | $\alpha$=0.1 | $\alpha$=0.3 | $\alpha$=0.5 |
| $T_c = 1$ | 50.72 | 59.41 | 63.89 | 54.344 | 79.87 | 80.14 |
| $T_c = 2$ | 63.08 | 65.90 | 71.16 | 56.13 | 79.75 | 85.18 |
| $T_c = 3$ | 61.61 | 69.73 | 73.91 | 74.41 | **86.42** | 86.18 |
| $T_c = 4$ | 66.26 | 69.40 | 74.39 | 78.67 | 86.36 | 86.43 |
| $T_c = 5$ | **67.65** | **71.42** | **76.01** | **80.28** | 86.25 | **86.55** |

DENSE improves as $T_c$ increases, and DENSE achieves the best performance when $T_c = 5$. This shows that DENSE can be extended to multi-round FL and the performance can be further enhanced by increasing the communication rounds.

### 3.3.5 Contribution of $\mathcal{L}_{BN}$ and $\mathcal{L}_{div}$

We investigate the contributions of different loss functions used in data generation. We conduct leave-one-out testing and show the results by removing $\mathcal{L}_{div}$ (w/o $\mathcal{L}_{div}$), and removing $\mathcal{L}_{BN}$ (w/o $\mathcal{L}_{BN}$). Additionally, we report the result by removing both $\mathcal{L}_{div}$ and $\mathcal{L}_{BN}$, *i.e.*, using only $\mathcal{L}_{CE}$ (w/ $\mathcal{L}_{CE}$). As illustrated in Table 6, using only $\mathcal{L}_{CE}$ to train the generator leads to poor performance. Be-

Table 6: Impact of loss functions in data generation.

| Dataset | CIFAR10 | SVHN | CIFAR100 |
|---|---|---|---|
| DENSE | **62.19** | **80.03** | **42.07** |
| w/ $\mathcal{L}_{CE}$ | 53.12 | 73.11 | 36.47 |
| w/o $\mathcal{L}_{BN}$ | 61.05 | 78.36 | 39.89 |
| w/o $\mathcal{L}_{div}$ | 59.18 | 77.59 | 39.14 |

sides, removing either the $\mathcal{L}_{BN}$ loss or $\mathcal{L}_{div}$ loss also affects the accuracy of the global model. A combination of these loss functions leads to a high performance of global model, which shows that each part of the loss function plays an important role in enhancing the generator.

## 4 Conclusion

In this paper, we propose an effective one-shot federated learning method called DENSE, which trains the global model by a data generation stage and a model distillation stage. Extensive experiments across various settings validate the efficacy of our method. Overall, DENSE is by far the most practical framework that can conduct data-free one-shot FL with model heterogeneity. A promising future direction is to consider the potential privacy attacks in one-shot FL.

## 5 Acknowledgement

This work was supported by Sony AI, the National Key Research and Development Project of China (2021ZD0110400 No. 2018AAA0101900), National Natural Science Foundation of China (U19B2042), Zhejiang Lab (2021KE0AC02), Academy Of Social Governance Zhejiang University, Fundamental Research Funds for the Central Universities.

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
