# 1 Appendix

## 1.1 Preliminaries and Related Works

### 1.1.1 Federated Learning

Suppose there are $m$ clients in a FL system, and each client $k$ has its own dataset $\mathcal{D}_k = \{\mathbf{x}_i, \mathbf{y}_i\}_{i=1}^{n_k}$ with $n_k = |\mathcal{D}_k|$ being the size of local data. Each client $k$ optimizes its local model by minimizing the following objective function,

$$\min_{\boldsymbol{\theta}^k} \frac{1}{n_k} \sum_{i=1}^{n_k} \mathcal{L}(f^k(\mathbf{x}_i), \mathbf{y}_i), \tag{1}$$

where $f^k()$ and $\boldsymbol{\theta}^k$ are the local model and model parameter, respectively. Then, client $k$ uploads $\boldsymbol{\theta}^k$ to the server for aggregation. After receiving the uploaded model parameters from the $m$ clients, the server computes the aggregated global model parameter as follows: $\boldsymbol{\theta}_S = \sum_{k=1}^{m} \frac{n_k}{n} \boldsymbol{\theta}^k$, where $n = \sum_{k=1}^{m} n_k$ is the total amount of training data (of all clients) and $\boldsymbol{\theta}_S$ is the server's global model parameter. Afterwards, the server distributes $\boldsymbol{\theta}_S$ to all clients for training in the next round. However, such a training procedure needs frequent communication between the clients and the server, thus incurs a high communication cost, which may be intolerable in practice [7].

### 1.1.2 One-shot Federated Learning

A promising solution to reduce the communication cost in FL is one-shot FL, which is first introduced by [3]. In one-shot FL, each client only uploads its local model parameter to the server once. After obtaining the global model, the server does not need to distribute the global model to the clients for further training. There is only one unidirectional communication between clients and the server, thus it highly reduces the communication cost and makes it more practical in reality. Moreover, one-shot FL also reduces the risk of being attacked, since the communication happens only once. However, the main problem of one-shot FL is the difficult convergence of the global model and it is hard to achieve a satisfactory performance especially when the data on clients are not independent and identically distributed (non-IID).

Guha *et al.* [3] and Li *et al.* [7] used ensemble distillation to improve the performance of one-shot FL. However, they introduced a public dataset to enhance training, which is not practical. In addition to model distillation, dataset distillation [11] is also a prevailing approach. Zhou *et al.* [13] and [5] proposed to apply dataset distillation to one-shot FL, where each client distills its private dataset and transmits distilled data to the server. However, data distillation methods fail to offer satisfactory performance compared with model distillation and sending distilled data can cause additional communication cost and potential privacy leakage. Dennis *et al.* [2] utilized cluster-based method in one-shot FL, but they required to upload the cluster means to the server, which incurs additional communication cost.

Overall, none of the above methods can be practically applied. In addition, none of these studies consider model heterogeneity, which is a main challenge in FL [8]. This leads to a fundamental yet so far unresolved question: *"Is it possible to conduct one-shot FL without the need to share additional information or rely on any auxiliary dataset, while making it compatible with model heterogeneity?"*

### 1.1.3 Knowledge Distillation in FL

In traditional FL frameworks, all users have to agree on the specific architecture of the global model. To support model heterogeneity, Li *et al.* [6] proposed a new federated learning framework that enables participants to independently design their models by knowledge distillation [4]. With the use of a proxy dataset, knowledge distillation alleviates the model drift issue induced by non-IID data. However, the requirement of proxy data renders such a method impractical for many applications, since a carefully designed dataset is not always available on the server.

Data-free knowledge distillation is a promising approach, which can transfer knowledge of a teacher model to a student model without any real data [1, 12]. Lin *et al.* [9] proposed data-free ensemble distillation for model fusion through synthetic data in each communication round, which requires high communication costs and computational costs. However, in this paper, we are more concerned with

obtaining a good global model through only one round of communication in cases of heterogeneous models, which is more challenging and practical.

Zhu *et al.* [14] also proposed a data-free knowledge distillation approach for FL, which learns a generator derived from the prediction of local models. However, the learned generator is later broadcasted to all clients, and then clients need to send their generators to the server, which increases the communication burden. More seriously, the generator has direct access to the local data (the generator can easily remember the training samples [10]), which can cause privacy concerns. As the generator used in our method is always stored in the central server, it never sees any real local data.