# OpenReview forum: "DENSE: Data-Free One-Shot Federated Learning"
_NeurIPS.cc/2022/Conference — NeurIPS 2022 Accept_

### Official Review · Reviewer_oqiW · 2022-06-25

**Rating:** 8
**Confidence:** 4
**Soundness:** 3 good
**Presentation:** 3 good
**Contribution:** 4 excellent

**Summary:**

The authors proposed a novel one-shot data-free Federated Learning algorithm:
1. The algorithm only requires a single communication rounds. The clients locally train the model and uploads the model to the central server.
2. In the server, the ensemble of the client model is utilized to train a generator model that generates synthetic data. The generator is trained by giving a random number and a random label, such that the logits computed from the generated image by averaging the client models minimize the cross entropy.
3. Then the generator is used to generate a large amount of synthetic data to be labeled by the client ensemble. One then distill the knowledge of the ensemble to a global model by train it with synthetic data.
4. The authors also includes additional loss terms for generator training to ensure similarity, stability, transferability of the data.
5. Finally, extensive experiments are carried out, comparing with existing Federated learning algorithm as well as ablation study.

**Questions:**

There are a few questions regarding the technical details:

1. The generator G(z) is trained by first sample random pairs of random number and label (z, y). Then the generated input x = G(z) is used such that x would minimize some loss. The author first shows cross entropy loss alone does not work well and conjecture that it is due to non-IID data. However, in ablation study one could certainly make the data IID for all clients (table 6). How does table 6 obtain the results? What is the hyperparameter used?

In my opinion the poor performance of cross entropy alone could also due to the sampling setup: we generate a random pair (z, y). There may not exist a proper transformation between z to x such that (z, y) is independent. In other words, (z, y) could be correlated.

2. How does the author handle the case that for client models, their logits distributions are very different (line 105-106)? In extreme case, suppose one model produce a set of logits of value O(1e6) while other models produce logits of value (1e-6). Then one model could clearly dominate over the average of the ensemble. Is this a concern for the averaging?

3. From algorithm 1, in the inner loop where generator is being updated, the global model's parameter theta_S is fixed. This would affect the quality of loss terms in equation (5) because the global model is still bad. Could the author comment on this issue?

4. In global model training (distillation), the author uses KL divergence as objective (equation (6)). Have the authors tried other losses such as cross entropy?



**Limitations:**

The author has described limitations of current work and suggest for future investigation. They have not discussed negative societal impact, which I believe is OK.

**Strengths And Weaknesses:**

The paper shows strength in the following perspectives:
1. Originality. The algorithm proposed is novel to my best knowledge. It addresses several issues of applying federated learning to real world applications. First, the algorithm is genuinely data free: no client data, distillation data or other data summary will be uploaded to server. Second, the algorithm admits heterogeneity of client models, such that in real world applications clients have the freedom of picking local models. Thirdly, the algorithm is one-shot in the sense that only one round of communication is needed, although the author mentioned that more rounds can improve the quality of the model.
2. Quality. The experiments conducted in the paper are extensive and sound. The author explains why some of the comparisons are not included such as regularization based methods. There are also plenty of experiments with a wide range of parameter space to demonstrate the effectiveness of the algorithm. Those include balancedness of the client data, number of clients, number of rounds and effects of added terms.
3. Clarity. The paper is well-written. The author reviews the existing limitations of the literature, and gives a clear structure on the components of the algorithm.
4. Significance. The paper's result is important both from academic and practical point of view. For the latter, it will stimulate the research community for real world application of FL, such that the whole infrastructure is secure, privacy-preserving, and high quality.

The weakness of the paper, in my opinion, has the following aspects:
1. Quality. The paper does not have a lot of theoretical analysis, especially on why the algorithm would converge regardless of which dataset one is using, and data distributions among clients, etc.
2. Clarity. While it is OK to move the backgrounds and preliminaries in appendix, in my opinion it would be better to move the full algorithm (Algorithm 1) in the main context - readers may constantly go to the table and refer to it.

---

> ### Author Response · Authors · 2022-08-02
> **Response to Reviewer oqiW**
>
>
> Thank you very much for your very detailed and supportive comments! We indeed highly appreciate your in-depth thought and discussion about our paper.
>
> > move the full algorithm (Algorithm 1) in the main context
>
> Thanks for your valuable suggestion. We will move Algorithm 1 to the main text in our final version.
>
> > How does table 6 obtain the results? What is the hyperparameter used?
>
> In the main text(line 240), we described that the default setting is $\alpha=0.5$ for non-IID settings, and Table 6 shows the experimental results when $\alpha=0.5$ on three datasets. Then we report the contributions of different loss functions used in our method. And yes, we also believe that $z$ and $y$ are somehow related. Here we use the generator to learn a proper transformation.
>
> > How does the author handle the case that for client models, their logits distributions are very different?
>
> Thanks for pointing this out. In non-IID FL, we think it is likely to be a case of model overfitting when a certain model makes an overconfident judgment. For example, when there are too many samples in class 0 and too few samples in class 1, the local model is likely to classify the test samples belonging to class 1 as class 0.  In this way, our ensemble method have  been  shown to yield robust measures of uncertainty, and are capable of distinguishing between different forms of uncertainty. And averaging model logits provides a simple and effective solution.
>
>
> > From algorithm 1, in the inner loop where generator is being updated, the global model's parameter theta_S is fixed. This would affect the quality of loss terms in equation (5) because the global model is still bad
>
> Yes, in the early stages of our method, the generator performance was not satisfactory and the student model did not converge. By alternately training the generator and student models, the two models will gradually become more accurate. The idea is somehow similar to adversarial training in GAN. We hope that the student model will be able to learn useful information from synthetic data and ensemble models.
>
> > In global model training (distillation), the author uses KL divergence as objective (equation (6)). Have the authors tried other losses such as cross entropy?
>
> Thank you for your question. As distillation loss is not our main contribution, we did not devote much time to tuning the distillation loss function. According to some studies[1,2], when performing model distillation, using the KL loss function to constrain the output of each model's logits is more effective than using the cross entropy loss. It is because the logits  contain more information than the one-hot label. Those[3,4] who use cross entropy loss for distillation often use a temperature hyperparameter  to soften the output value of softmax to some extent. Due to this, we think that KL loss or the cross-entropy loss function with temperature parameters should be suitable for model distillation. Here are experiments on CIFAR10 and SVHN.
>
> | Loss | KL loss |  Cross-Entropy loss|
> | :----: | :----:  | :----: |
> | CIFAR10 | 62.56 | 60.13|
> |SVHN|  79.64  |  77.83     |
>
> [1] Zhang, Jie, et al. "QEKD: Query-Efficient and Data-Free Knowledge Distillation from Black-box Models." arXiv preprint arXiv:2205.11158 (2022).
>
> [2] Truong, Jean-Baptiste, et al. "Data-free model extraction." Proceedings of the IEEE/CVF Conference on Computer Vision and Pattern Recognition. 2021.
>
> [3] Nayak, Gaurav Kumar, et al. "Zero-shot knowledge distillation in deep networks." International Conference on Machine Learning. PMLR, 2019.
>
> [4] Hinton, Geoffrey, Oriol Vinyals, and Jeff Dean. "Distilling the knowledge in a neural network." arXiv preprint arXiv:1503.02531 2.7 (2015).
>
> > The paper does not have a lot of theoretical analysis
>
> Thank you for your valuable suggestion, and we also believe that adding theoretical analysis can further enhance the quality of our paper. Just as you mentioned, in the area of training with synthetic data, especially the data-free setting, the development of practice is usually ahead of theory. Advancing the theoretical progress in this field is a valuable future research direction.

---

### Official Review · Reviewer_fNZT · 2022-07-07

**Rating:** 7
**Confidence:** 4
**Soundness:** 3 good
**Presentation:** 3 good
**Contribution:** 3 good

**Summary:**

This paper proposed a novel data-free one-shot FL framework named DENSE based on data generation and knowledge distillation techniques so that it can be applied to communication-efficient heterogeneous FL. Evaluation results on multiple benchmark datasets illustrate the good performance of the proposed FL framework.

**Questions:**

Why not compare the proposed method with the FedGen method that is data-free FL with knowledge distillation? I know it requires to send the model parameters of generator between server and clients multiple rounds.

**Limitations:**

Yes, the authors have described the limitations of this work.

**Strengths And Weaknesses:**

Strengths:
[1] This paper proposed a novel data-free one-shot FL framework consisting of two stages: i) trains a generator based on ensemble models uploaded from clients and random labels; ii) adopts knowledge distillation to transfer the output from the teacher model to the global model. The idea of data-free one-shot FL is very interesting.
[2] Extensive experiments have been conducted to evaluate the performance of the proposed framework

Weakness:
[1] I am concerned about the privacy as the ensemble models is uploaded to the central sever. As far as I know, attackers in the server side can still recover the privacy data based on the model parameters.
[2] I am wondering if the labelled data in the server would leak the data privacy? Such as race and gender.
[3] Please explain the important steps in algorithm 1 in Appendix with mode details. It seems hard to understand the algorithm.
[4] Please conduct experiments with mutiple random seeds, and then report the mean and variance. Otherwise, the experimental results seem not very convincing.

---

> ### Author Response · Authors · 2022-08-02
> **Response to Reviewer fNZT, Part 1**
>
> We thank Reviewer fNZT for the careful review. We clarify your points mentioned in the comments as follows.
>
> > Attackers in the server side can still recover the privacy data based on the model parameters.
>
> As reviewer e2Cu said, "one-shot FL is already more secure than multi-round FL". We would like to emphasize that our method is more secure than multi-round FL. The reasons are as follows:
>
> - Assumes the client is malicious: Compared to multi-round FL, our method greatly mitigates privacy and security risks during communication. In multi-round FL, attackers can continuously modify the data (data poisoning) or model (model update poisoning:) to modify the behavior of the model in some undesirable way. However, if there is only one round of communication, an attacker would have difficulty launching a successful attack.
>
> - Assumes the server is malicious: Yes, some studies[1,2] have shown that even without any real training data, attackers can still recover the data through model parameters. Imagine attacking a well-trained model (84% accuracy ) trained in multi-round FL and a model (63% accuracy) trained in one-shot FL. It is easier for attackers to recover private information from a better model. Thus, we believe that once the central server becomes malicious, the attacker can recover the private data more easily in multi-round FL than in one-shot FL.
>
> In light of the above discussion, we believe that our approach does not pose any additional privacy concerns in comparison with multi-round FL.
>
> Furthermore, several existing privacy-preserving methods can be incorporated into our framework to protect clients from adversaries. We leave this as our future work.
>
> [1] Yin H, Molchanov P, Alvarez J M, et al. "Dreaming to distill: Data-free knowledge transfer via deepinversion", CVPR 2020.
>
> [2] Yin H, Mallya A, Vahdat A, et al. "See through gradients: Image batch recovery via gradinversion" , CVPR 2021.
>
> ---
>
> > if the labelled data in the server would leak the data privacy?
>
> We visualize the generated data in Figure 6 (learn from the models pretrained on CIFAR10 and SVHN datasets ). Clearly, the synthetic data are not similar to the original data, which can effectively reduce the probability of leaking sensitive information of clients.
>
> ---
>
> > Why not compare the proposed method with the FedGen method that is data-free FL with knowledge distillation?
>
> Thanks  for your suggestion!  First, we would like to emphasize that FedGen[2] needs to broadcast the generator parameter in each communication round, which means it heavily relies on frequent communication to continuously regulate the local training.  It is also d
>  ef ense to note that FedGen provides results only on simple datasets, such as MNIST and EMNIST. However, our approach can be adapted to more complex datasets, such as tiny-imagenet and CIFAR100.
>
> Based on your  suggestion,  we try our best to compare FedGen with our method on the same configuration. We conducted experiments on CIFAR10 and MNIST datasets with $\alpha$=0.1. Below is a comparison of the two methods in the one-shot FL:
>
> | Method | MNIST | CIFAR10 |
> | :----: | :----:  | :----: |
> | FedGen | $51.32_{\pm 1.62}$ | $28.31_{\pm1.93}$ |
> | Ours | $66.57_{\pm1.31}$ | $50.31_{\pm1.56}$ |
>
> We observed poor performance while applying FedGen on CIFAR10 dataset. Several researchers have also found that FedGen performs poorly on CIFAR10 (see detailed issues in FedGen's github code link). We hope these new results have addressed your concerns.
>
> ---
>
> > Please conduct experiments with mutiple random seeds
>
> Thanks for your valueable comments. The following table shows the results on CIFAR10 and CIFAR100 ($\alpha=0.5$) using 10 random seeds and reports their average and standard deviation.
>
> | Method | Fed-DAFL | Fed-ADI | Ours |
> | :----: | :----:  | :----: | :----: |
> |CIFAR10|	$58.52_{\pm 1.37}$ |	$59.31_{\pm1.21}$|	$63.06_{\pm 1.32}$|
> |CIFAR100|	$38.34_{\pm2.03}$|	$40.06_{\pm0.95}$|	$42.56_{\pm1.41}$|
>
> We hope these new results have addressed your concerns about the experimental results.

---

> ### Author Response · Authors · 2022-08-02
> **Response to Reviewer fNZT, Part 2**
>
> > Please explain the important steps in algorithm 1
>
> Sorry for the confusion. Following is a detailed description of our method.
> 1) At first, the server side will collect $n$ local models that have been trained to converge (e.g., for 200 epochs). Then ensemble these $n$ models as a teacher model $T$. Randomly initializes a generator $G$ and a student model $S$.
> 2) Afterwards, train the generator $G$ and the student model $S$ alternately on the server.
> We will quickly cleaned our code, benchmark datasets and pre-trained model. Our code will soon be available.  We attach our key implementation of DENSE as below.
>
> ```python
> def ensembel_distill(teacher, student, generator):
>     syn_data = generator.get_saved_data()
>     with tqdm(syn_data) as epochs:
>         for idx, (syn_images) in enumerate(epochs):
>             t_out = teacher(syn_images)
>             s_out = student(syn_images.detach())
>             distill_loss = KLDiv(s_out, t_out.detach())
>
>
> def generate_data(self, teacher, student, generator):
>     hooks = []
>     for m in teacher.modules():
>         if isinstance(m, nn.BatchNorm2d):
>             hooks.append(Hook(m))
>     with tqdm(total=self.iterations) as t:
>         for it in range(self.iterations):
>             inputs = generator(z)
>             t_out = teacher(inputs)
>             loss_bn = sum([h.r_feature for h in hooks])
>             loss_oh = F.cross_entropy(t_out, targets)
>             s_out = student(inputs)
>             mask = (s_out.max(1)[1] != t_out.max(1)[1]).float()
>             loss_adv = -(KLDiv(s_out, t_out, reduction='none').sum(
>                 1) * mask).mean()
>             loss = self.bn * loss_bn + self.oh * loss_oh + self.adv * loss_adv
>
> for epoch in range(epochs):
>     # 1. Data generation
>     generate_data(teacher, student, generator)
>     # 2. Ensemble distillation
>     ensembel_distill(generator, student, teacher)
> ```

---

> > ### Comment · Reviewer_fNZT · 2022-08-03
> > **Use any labelled data to train the ensemble model on the server?**
> >
> > Thanks lot for your response! I am curious if you need to use any labelled data to train the ensemble on the server? If it does, I am concerned the data privacy issue.

---

> > > ### Author Response · Authors · 2022-08-03
> > > **Thanks for your quick feedback**
> > >
> > > > if you need to use any labelled data to train the ensemble on the server
> > >
> > > Thank you very much for your quick feedback! In fact, as you can see from the code above, we do not require any labeled data for both data generation and model distillation. In our method, the ensemble model is always frozen, and only the **unlabeled synthetic data** is used to train the student model. We hope these new discussion have addressed your concerns about the data privacy issue.
> > >
> > > Again, we highly appreciate knowing if our responses have addressed your questions. We are delighted to answer your remaining concern. We appreciate your inputs and feedback very much. Thank you!

---

> ### Author Response · Authors · 2022-08-06
> **Thank you for your comments**
>
> Dear Reviewer fNZT,
>
> Thank you again for your support of our work and valuable feedback!  We tried our best to address all mentioned concerns/problems. Are there unclear explanations? We could further clarify them. Could you please kindly re-evaluate our paper based on the current situation? If you have any further questions, we are very glad to discuss them.
>
> We highly appreciate knowing if our responses have addressed your initial questions.
>
> Thank you a lot!

---

> > ### Comment · Reviewer_fNZT · 2022-08-06
> > **I have modified my score**
> >
> > Hi
> >
> > Thank you for your detailed response and I have improved your score.

---

### Official Review · Reviewer_e2Cu · 2022-07-10

**Rating:** 8
**Confidence:** 5
**Soundness:** 3 good
**Presentation:** 4 excellent
**Contribution:** 4 excellent

**Summary:**

The paper focuses on one-shot federated learning, i.e., the server can learn a model with a single communication round. The proposed FedSyn method has two stages: first, training a generator from the ensemble of models from clients; second, distilling the knowledge of the ensemble into a global model with synthetic data. The authors validate the efficacy of FedSyn by conducting extensive experiments on 6 different datasets with various non-IID settings generated from Dirichlet distributions. Results can well support that the proposed method consistently outperforms all the baselines.

**Questions:**

1. What is the architecture of GAN? Is a larger GAN better for model distillation?

2. In Fig.3, why the performance of Fedavg decreases with longer training epochs?

3. In Table 6, how about the results for "w/o l_CE"? Is l_CE necessary?

4. In the future direction, authors mentioned “defend privacy attacks in one-shot FL”, could you elaborate more on this? From my understanding, one-shot FL is already more secure than multi-round FL.


**Limitations:**

The analysis, theory, and method are sound to me, but I didn't check the privacy concern.

**Strengths And Weaknesses:**

Strengths
1. This paper focuses on one-shot FL, an interesting but less explored topic. From my understanding, the proposed method is by far the most practical one considering that：1) DENSE requires no additional information to be transferred between clients and the server; 2) DENSE does not require any auxiliary dataset for training; 3) DENSE considers both model heterogeneity, i.e., different clients with different model architectures. Generally, I think the investigated problem is sound and interesting. I think this can be an extremely strong paper in one-shot FL.

2. The method using data-free ensemble distillation is inspiring and novel. The experimental evaluation of applying the data-free ensemble distillation to one-shot FL significantly improves the performance of the global model. Fig. 1 illustrates the method clearly and is well drawn.

3. The experimental settings are well motivated, and the related analysis is convincing. My favorite parts of the paper are the discussion and the ablation study, which offer sound support for the proposed DENSE.

Overall, this paper is very interesting and to my knowledge novel. It seems like a pioneering contribution towards practical one-shot federated learning. Hence, I would like to vote for strong acceptance.

Weaknesses.

Below questions need to be addressed to further improve the quality of this paper.
1. What is the architecture of GAN? Is a larger GAN better for model distillation?

2. In Fig.3, why the performance of Fedavg decreases with longer training epochs?

3. In Table 6, how about the results for "w/o l_CE"? Is l_CE necessary?

4. Even though authors provide sufficient references, a few important papers are recommended. For example, a recent IJCAI paper “Data-Free Adversarial Knowledge Distillation for Graph Neural Networks” that also studied data-free distillation. These papers did not study exactly the same topic as this paper, but would certainly further enrich the literature review.

---

> ### Author Response · Authors · 2022-08-02
> **Response to Reviewer e2Cu**
>
> We thank Reviewer e2Cu for the very positive and constructive feedback. We indeed highly appreciate your in-depth comments and summary about our paper.
>
> > What is the architecture of GAN? Is a larger GAN better for model distillation?
>
> For fair comparisons, we use the same generator for all methods in our experiments. We also introduce the effects of different sizes of generators, as shown in the table, where DCGAN, StyleGAN and Transformer-GAN have small, medium and large parameters. Different generative models have relatively minor effect on the performance.
>
> | Generator | DCGAN | StyleGAN| Transformer-GAN |
> | :----: | :----:  | :----: | :----: |
> | Ours | 62.64 | 63.21 | 63.83 |
>
>
> > In Fig.3, why the performance of Fedavg decreases with longer training epochs?
>
> As shown in Fig.3, the global model achieves the best performance (test accuracy=34\%) when $E=40$, while a larger value of $E$ can cause the model to degrade even collapse. This result can be attributed to the inconsistent optimization objectives with non-IID data, which leads to weight divergence. Thus, it is not suitable to use FedAvg if there are too many local training rounds as the model parameters will fluctuate too much.
>
>
>
> > In Table 6, how about the results for "w/o l_CE"? Is l_CE necessary?
>
> Yes, the cross-entropy loss function is the most basic way to update the generator.
> If the cross-entropy loss function is removed, the image will appear more like random noise.
>
> > a few important papers are recommended
>
> Thanks for the valuable suggestion! In the final version, we will include these recent related papers.

---

### Official Review · Reviewer_4PjE · 2022-07-14

**Rating:** 4
**Confidence:** 5
**Soundness:** 3 good
**Presentation:** 2 fair
**Contribution:** 3 good

**Summary:**

This paper considers the problem of training a global model with only one-shot communication between the server and clients in the federated learning setting. This problem is addressed by designing a generator based on the ensemble of 'trained' local models, to generate synthetic data samples, which can then be further utilized to distill the ensemble models to the global model in a knowledge distillation paradigm.

**Questions:**



Please refer to item **'Weaknesses'** in the previous section.

**Strengths And Weaknesses:**

[Strengths]

- The one-shot federated learning is an important problem in both areas of machine learning and distributed computing.

- Using a generator trained from local models can provide synthetic data samples for the knowledge distillation between the ensemble models and the global model, which is also a simple solution for data-free distillation.

- The evaluation has been conducted on various datasets, which is good.

[Weaknesses]

- The novelty is limited. The setting of 'data-free distillation' + 'heterogeneous local models' has been already addressed by some previous works. For example, FEDGEN [1] also trains a generator to provide data-free knowledge distillation between the server and clients, wherein, their local models are also in heterogeneous architectures. The idea is almost the same.

- The one-shot relies heavily on the 'well-trained' local models, to reduce the communication overhead between the server and clients. However, such a setting is not so practical when the local devices are resource constrained, in either data or computing capability.

- Using the ensemble output results of uploaded local models may cause serious privacy issues. In that case, each individual local model can be examined on both input and output, thus the attackers can have the potential to reconstruct local models or local samples if the server is adversary-oriented.

- The writing and presentation should be improved.

[1] Data-Free Knowledge Distillation for Heterogeneous Federated Learning, ICML 2021.

---

> ### Author Response · Authors · 2022-08-02
> **Response to Reviewer 4PjE**
>
>
> We thank Reviewer 4PjE for the comments and summary of our paper. We have addressed all your questions in the following.
>
> > The novelty is limited.
>
>
> First, we want to point out that this paper is primarily focused on one-shot FL. Although some studies [1,2] have used data-free distillation in FL, these methods are impractical in one-shot FL. For example, [1] requires frequent communication with clients, and FedGen[2] needs to broadcast the generator parameter in each communication round, which means it heavily relies on frequent communication to continuously regulate the local training.  The dependency on frequent communication makes these methods exhibit poor performance in one-shot FL scenarios.
>
> Moreover, we provide a detailed comparison with FedGen[2] as follows:
>
> | Method | #Communication | Broadcast | Distillation|
> | :----: | :----:  | :----: | :----: |
> |FedGen|	multi-round	|predictor+generator|	At client side, use a generator  to generate data in local training|
> |Ours	|single round|	local model|	At server side, ensemble distillation with synthetic data|
>
>
> Essentially, FedGen[2] uses the generator to regulate local training. Thus FedGen[2] heavily relies on frequent communication to update the generator. But for our method, there is no need to send the generator to the client. Once the server has collected local models, all training can be completed on the server. It is also worthwhile to note that FedGen provides results only on simple datasets, such as MNIST and EMNIST. However, our approach can be adapted to more complex datasets, such as tiny-imagenet and CIFAR100.
>
> The above comparison can highlight the technical characteristics of our work compared to existing methods. We believe that the innovative use of data-free distillation can advance the field of one-shot FL.
>
> [1] Lin, Tao, et al. "Ensemble distillation for robust model fusion in federated learning." NeurIPS 2020.
>
> [2] Zhu, Zhuangdi, et al. "Data-free knowledge distillation for heterogeneous federated learning." ICML, 2021.
>
> ---
>
> > The one-shot relies heavily on the 'well-trained' local models.
>
>
> Thank you for pointing this out. “well-trained” refers to the locally trained model which has converged, not the model with particularly good performance. For example, the local model in the table below has only 35.21% accuracy, whereas our method obtains a model with 49.76% accuracy. In our method, users only have to train their own models based on their actual circumstances (e.g. , users with limited resources can design smaller models).  Moreover, as shown in Tables 2 and 3, our method outperforms the baseline methods in scenarios with small models and limited data.
>
> ---
>
> > The attackers can have the potential to reconstruct local models or local samples if the server is adversary-oriented.
>
>
> As reviewer e2Cu said, "one-shot FL is already more secure than multi-round FL". We would like to emphasize that our method is more secure than multi-round FL. The reasons are as follows:
>
> - Assumes the client is malicious: Compared to multi-round FL, our method greatly mitigates privacy and security risks during communication. In multi-round FL, attackers can continuously modify the data (data poisoning) or model (model update poisoning:) to modify the behavior of the model in some undesirable way. However, if there is only one round of communication, an attacker would have difficulty launching a successful attack.
>
> - Assumes the server is malicious: Yes, some studies[3,4] have shown that even without any real training data, attackers can still recover the data through model parameters. Imagine attacking a well-trained model (84% accuracy ) trained in multi-round FL and a model (63% accuracy) trained in one-shot FL. It is easier for attackers to recover private information from a better model. Thus, we believe that once the central server becomes malicious, the attacker can recover the private data more easily in multi-round FL than in one-shot FL.
>
> In light of the above discussion, we believe that our approach does not pose any additional privacy concerns in comparison with multi-round FL.
>
> Furthermore, several existing privacy-preserving methods can be incorporated into our framework to protect clients from adversaries. We leave this as our future work.
>
> Besides, we visualize the generated data in Figure 6 (learn from the models pretrained on CIFAR10 and SVHN datasets ). Clearly, the synthetic data are not similar to the original data, which can effectively reduce the probability of leaking sensitive information of clients.
>
> [3] Yin H, Molchanov P, Alvarez J M, et al. "Dreaming to distill: Data-free knowledge transfer via deepinversion", CVPR 2020.
>
> [4] Yin H, Mallya A, Vahdat A, et al. "See through gradients: Image batch recovery via gradinversion" , CVPR 2021.
>
> ---
>
> > The writing and presentation should be improved.
>
>
> Thanks for your advice. We will carefully revise it in the final version.

---

> > ### Comment · Area_Chair_7NjZ · 2022-08-03
> > **followup**
> >
> > Dear reviewer,
> > Can you please take a look at the author's reply and other reviews and see if they addressed your concerns in any way? Thanks.

---

> > > ### Comment · Reviewer_4PjE · 2022-08-07
> > > **score updated**
> > >
> > > i have updated my score, thanks.

---

> ### Author Response · Authors · 2022-08-06
> **Thank you for your comments**
>
> Dear Reviewer 4PjE,
>
> We would like to thank the reviewer for taking the time to review our paper and for the comments.
>
> We have now clarified the significance and the novelty of our method and also show a detailed comparison with FedGen. Note that more detailed information is shown in our rebuttal summary.
>
> Please kindly let us know if anything is unclear. We truly appreciate this opportunity to improve our work and shall be most grateful for any feedback you could give to us.
>
> Thanks again!

---

### Meta-Review · Area_Chair_7NjZ · 2022-08-24

**Recommendation:** Accept
**Confidence:** Less certain

**Metareview:**

This work proposes a new one-shot FL algorithm. It consists of two steps on the server: a data generation step that trains a GAN to synthesize data utilizing the local models and a distillation step that distills the ensembles local models using the generated data. The method has several advantages in comparison with other one-shot FL algorithms. The performance is verified by experiments.

One major concern in the reviews was regarding novelty. This has been addressed by the author.

Please clarify the following in the final version
1. Teacher's model: What is the quality of the ensemble model (teacher) in the experiment? Does the distilled model improve over the teacher (similar to self-distillation)? Showing the distillation gap is important to understand how the method works.
2. Contribution of GAN in quality: From pure quality point of view, what if the original data is used to train the ensemble and distilled models?

Please also consider adding privacy-utility trade offs in the future work. It is true that one-shot FL is in general more secure than multi-round methods, and some DP work can be applied here directly. But showing on-par or better privacy-utility trade off is an important justification on why it should be adopted.

**Award:**

No

---

### Decision · Program_Chairs · 2022-09-14

Accept